# Dynamic Quantitative Imaging of the Masseter Muscles in Bruxism Patients with Myofascial Pain: Could It Be an Objective Biomarker?

**DOI:** 10.3390/jpm13101467

**Published:** 2023-10-06

**Authors:** Sibel Aydin Aksu, Pinar Kursoglu, Izim Turker, Fulya Baskak, Elifnaz Ozen Sutuven, Kaan Meric, Fatih Cabbar

**Affiliations:** 1Department of Radiology, Haydarpasa Numune Training and Research Hospital, University of Health Sciences, 34668 Istanbul, Turkey; fulyabaskak@hotmail.com; 2Department of Prosthodontics, Yeditepe University Faculty of Dentistry, 34728 Istanbul, Turkey; pinar.kursoglu@yeditepe.edu.tr (P.K.); elifnaz.ozen@yeditepe.edu.tr (E.O.S.); 3Department of Prosthodontics, Bahcesehir University School of Dental Medicine, 34357 Istanbul, Turkey; izim.turker@bau.edu.tr; 4Department of Medical Imaging Techniques, Beykoz University, 34805 Istanbul, Turkey; kaanmeric@beykoz.edu.tr; 5Department of Oral and Maxillofacial Surgery, Yeditepe University Faculty of Dentistry, 34728 Istanbul, Turkey; fcabbar@gmail.com

**Keywords:** shear wave elastosonography, ultrasonography, bruxism, myofascial pain, masseter muscle, thickness, stiffness, occlusal splint

## Abstract

We aimed to investigate whether the collaboration of shear wave elastosonography (SWE) and B-mode ultrasonography (US) could be offered as diagnostic tools to assess the presence, severity, and progress of bruxism, as well as a biomarker for the effectiveness of treatment in daily clinical practice. The study was designed as a quantitative evaluation of the masseter muscles (MMs) of the clinically diagnosed bruxism patients suffering from myofascial pain and MMs of the healthy individuals. Clinical examinations were made according to the diagnostic criteria for temporomandibular disorders (DC/TMD), and pain was assessed using a visual analog scale (VAS). Painful MMs with VAS scores ≥ 4 were assigned to Group A, and healthy MMs were assigned to Group B. Also, the MMs of the painful bruxers were analyzed based on wearing occlusal splints. Group A was divided into two subgroups as splint users (Group AI) and non-users (Group AII). All the participants were scanned with dynamic US and SWE to quantify the size and stiffness of the MMs. Measurements of each muscle pair while the jaw is in a resting position (relaxation) and clenching position (contraction) were recorded. The significant differences in stiffness and thickness became visible in the relaxation state. Bruxism patients with myofascial pain had significantly harder and thinner MMs than healthy individuals. During the relaxation, the mean thickness and elasticity values were 9.17 ± 0.40 mm and 39.13 ± 4.52 kPa for Group A and 10.38 ± 0.27 and 27.73 ± 1.92 for Group B, respectively. Also, stiffer MMs were measured in Group AII (38.16 ± 3.61 kPa) than in Group AI (26.91 ± 2.13 kPa). In conclusion, the combination of SWE and US using a dynamic examination technique has the potential to be a valuable tool for the management of bruxism patients suffering from myofascial pain.

## 1. Introduction

Bruxism is a stomatognathic system dysfunction of great attention to both researchers and clinicians in the dental, neurological, imaging, and sleep medicine areas with more than 1800 articles published in MEDLINE, including 259 review papers in the last 10 years [1]. It is defined as a repetitive jaw muscle activity characterized by clenching or grinding of the teeth and/or by bracing or thrusting of the mandible and specified as either sleep bruxism or awake bruxism depending on its circadian manifestation [2].

The clinical relevance of bruxism is linked to its destructive influence on the masticatory system. This diurnal or nocturnal parafunctional activity may lead to masticatory muscle hypertrophy or atrophy, stiffness, and fatigue in jaw muscles, resulting in impaired coordination and restrictions of jaw movements, myofascial pain, and damage to dental hard tissues. These conditions make it essential to accurately assess the presence and extent of bruxism in daily clinical practice [3].

Bruxism is usually considered to be one of the leading causes of TMD disorders. The available data demonstrate a positive relationship between bruxism and TMD, with bruxism increasing the risk of developing TMD later in life [4]. According to the diagnostic criteria for TMD (DC/TMD) Axis I, TMD could be divided into three as Group I: muscle disorders (including myofascial pain with and without mouth opening limitation); Group II: disc displacements with or without reduction and mouth opening limitation; and Group III: arthralgia, arthritis, and arthrosis [5]. It should be considered that bruxism may cause TMD associated with myofascial pain.

The diagnosis is based on the consequences of bruxism and mainly depends on a subjective assessment conducted by patients and clinicians; therefore, the accuracy of such an assessment is not sufficient [2,6].

A recent consensus approach for the multidimensional evaluation of bruxism status, comorbid conditions, etiology, and consequences was revealed as the standardized tool for the assessment of bruxism (STAB). This instrument includes self-reported information on bruxism status and possible patient consequences by self-reports and a clinical and instrumental assessment [7].

Several suggestions for the assessment of bruxism were put forward during the international consensus meeting in San Francisco. To this end, the A4 principle was advised for researchers to use: accurate (reliable, valid), applicable (feasible), affordable (cost-effective), and accessible (suitable for everyday clinical use) [6].

In that context, ultrasound imaging techniques stand out among others by offering a real-time and dynamic evaluation of the masseter muscle (MM). It provides greater sensitivity for deeper structures and better spatial resolution and allows one to obtain information about muscle size, shape, and muscle movement during specific tasks or functional activities. It can also capture the dynamic changes in muscle activity during jaw movements. Moreover, the advantage of performing B-mode ultrasonography (US) and shear wave elastosonography (SWE) on the same device makes it more feasible, as well as its easy application, low cost, high accessibility, reproducibility, as well as its lack of exposure to ionizing radiation [8].

SWE is an up-to-date imaging modality that facilitates the evaluation of soft tissue viscoelasticity by quantifying the hardness. In the context of bruxism, it can be used to evaluate the stiffness of the MM, which is the primary muscle involved in jaw clenching and grinding. In contrast to the strain elastosonography, SWE has an advantage in terms of higher reproducibility and quantification of the stiffness that can be expressed in kPa or meters/second [9,10].

In this study, we quantified MM stiffness and size in the course of certain tasks to question whether SWE and US combination could be offered as diagnostic tools to assess and manage bruxism patients with myofascial pain. We also investigated alterations due to the use of occlusal splints and if these measurements could be a biomarker for the effectiveness of treatment in daily clinical practice.

## 2. Materials and Methods

### 2.1. Ethical Consideration

The study was conducted following the principles of the Helsinki Declaration, and the study protocol was approved by the Clinical Research Ethics Committee of Yeditepe University Faculty of Dentistry (file date and number: 2019/942). All participants were informed of the purpose of the study and signed a written consent form.

### 2.2. Study Design

This study was designed to investigate if quantitative imaging methods could be used as an objective tool for the assessment and monitoring of bruxism patients with myofascial pain.

Clinical examinations were performed according to the diagnostic criteria for temporomandibular disorders (DC/TMD) by one experienced practitioner in this field. Pain due to spasm was examined by the palpation of the MM, and this was assessed using a visual analog scale (VAS) according to the DC/TMD Axis I [5]. Its intensity was scored by the subjects from 0 to 10 on the scale. Bruxism patients were selected according to a subject-based assessment (self-report) and clinically based assessment (examiner report) of STAB [7].

The study population was mainly consisted of two groups. Group A included painful MMs of bruxism patients with VAS scores ≥ 4. An equal number of healthy MMs in similar age and sex were selected as the control group and referred to as Group B. To analyze the efficacy of the occlusal splints in painful bruxer patients, we divided Group A into two subgroups as splint users (Group AI) and non-users (Group AII).

All the MMs were scanned with dynamic B-mode US and SWE. Measurements of each muscle pair while the jaw is at resting position (relaxation) and clenching position (contraction) were recorded. MM thickness and width were measured for evaluating hypertrophy or atrophy (mm) and stiffness of the muscles indicating spasm (kPa).

Subjects who had neurological diseases, pathologies such as tumors or inflammation, botulinum toxin injections in the examined area, medications that could interfere with muscle activities, and those under analgesic/anti-inflammatory treatment were excluded.

### 2.3. Imaging and Measurements

MM antero-posterior (thickness) and transverse (width) dimensions and shear wave velocity (Vs) measurements were performed in a half-dark examination room in the supine position by one experienced radiologist who was blinded to the diagnosis. An Applio 500 diagnostic color-Doppler ultrasound system (Toshiba Medical Systems Corporation, Tochigi, Japan) with a 17,5 MHz linear transducer was used. A water-based transmission gel was applied before the procedure.

The bilateral MMs were scanned by placing the probe on a line between the mouth angle and the lower implantation of the ear, crossing over the muscle’s transverse plane, approximately corresponding to the bulkiest part of it. During imaging, the transducer was held perpendicular, and special care was taken to avoid excessive pressure (Figure 1).

Real-time dynamic imaging of the muscle was obtained at rest and during jaw clenching with maximum force. Thickness was measured at three points by dividing it into three even parts in the transverse plane (anterior, middle, and posterior thirds), each at contraction and relaxation status. The width was measured as well. Measurements were repeated three times for each point, and mean values were recorded both at resting and clenching positions (Figure 2).

Quantitative evaluation of MM stiffness was performed using SWE, which works based on the physical fact that some of the energy of the ultrasonic waves is converted into force that moves the object away from the ultrasound source. So, a higher vs. value indicates greater stiffness. By making ROI’s (region of interest) the anterior, middle, and posterior thirds, each MM was scanned at rest and during jaw clenching with maximum force (Figure 3). The shear wave images were obtained in “one shot scan” mode, in which image quality is given higher priority. Stiffness is expressed in the elasticity mode in kilopascals (kPa), (range, 0–180 kPa). The system automatically calculated the mean elasticity (E_mean_), maximum elasticity (E_max_), and standard deviation of the elasticity (SD). A semi-transparent 2D-colour SWE map was obtained which displays the distribution of stiffness by the gradual colors, with increasing stiffness revealed in ascending order of blue, green, yellow, and red being obtained. Propagation mode was used to verify the reliability of the acquired data (Figure 4). Each measurement was repeated three times under each condition, and the three measurements were averaged.

### 2.4. Statistical Analysis

Data were statistically analyzed with the IBM SPSS^®^ Statistics v.22 (IBM SPSS Inc., Chicago, IL, USA). Before analyzing the relationship of continuous numerical variables between groups, normality tests were performed considering the number of samples in the groups. Normality was checked using the Shapiro–Wilk test. Accordingly, normally distributed variables were reported as mean ± standard deviation (SD). Student’s *t*-test, in which the mean values were compared for those with normal distribution, was used to analyze the difference between the two groups in terms of numerical variables. *p*-values less than 0.05 were considered significant.

## 3. Results

In total, 52 MMs of 26 subjects were included in the study, of which 19 (73.1%) were female and 7 (26.9%) were male. The age ranged from 20 to 57, with a mean age and SD of 28.0 ± 9.98 years. Group A consisted of 26 painful MMs of bruxism patients with VAS scores ≥ 4. An equal number of healthy MMs were included as the control group and referred to as Group B. To analyze the usefulness of the quantitative imaging methods in the field of treatment monitoring, the painful bruxer group was divided into two subgroups as splint users (Group AI) and non-users (Group AII). The MMs of Group AI (*n* = 14) and Group AII (*n* = 12) were compared.

In terms of the evaluation of the MM biometry, MM width and thickness measurements were obtained using B-mode dynamic ultrasound. For all participants, the mean width values of MMs were significantly higher in relaxation status than in contraction; however, the thickness values were significantly lower (Table 1).

In the comparison of Group A and Group B, the mean thickness values and SDs were 9.17 ± 0.40 mm and 10.97 ± 0.44 mm for Group A and 10.38 ± 0.27 mm and 11.98 ± 0.32 mm for Group B in the state of relaxation and contraction, respectively. There was a statistically significant difference between the two groups (*p* = 0.016) in relaxation status, but this was statistically insignificant (*p* = 0.081) in contraction. The mean transverse diameters and SDs were 44.08 ± 1.24 mm and 43.57 ± 1.52 for Group A and 50.55 ± 1.35 mm and 48.83 ± 1.64 for Group B in the state of relaxation and contraction, respectively. The difference between the two groups was found to be statistically significant under each condition (Table 2).

In terms of stiffness measurement, the mean elasticity values, and SDs of the MMs at rest were as follows: 39.13 ± 4.52 kPa for Group A and 27.73 ± 1.92 kPa for Group B, which were significantly higher in patients with bruxism than in the controls. In contraction status, the mean elasticity measurements were 44.04 ± 4.99 kPa for Group A and 35.52 ± 2.64 kPa for Group B. The difference between the two groups was statistically significant for MM stiffness during relaxation (*p =* 0.008) but not during contraction (*p =* 0.105).

The relationship between the use of occlusal splints and MM stiffness in painful bruxers was also examined. In the state of relaxation, the mean elasticity and SD values were 26.91 ± 2.13 kPa for Group AI and 38.16 ± 3.61 kPa for Group AII. The *p*-value for the difference was less than 0.01, indicating that it was statistically significant during relaxation (*p* = 0.006). But in the contraction state, the values were 37.21 ± 3.06 kPa for Group AI and 39.64 ± 4.03 kPa for Group AII. The difference between the two groups was 2.43 kPa (CI 7.63–12.50), and this was statistically insignificant (*p* = 0.629).

Dimensions of the MMs in painful bruxers were also compared based on occlusal splint usage. It was found that there were no statistically significant differences in MM thickness and width between the two groups with *p*-values higher than 0.10 (Table 3).

## 4. Discussion

It is estimated that bruxism affects around 8–31% of the general population. However, the exact prevalence is difficult to determine because the absence of standardized diagnostic criteria and the fact that only around 15% of people with bruxism are aware of their condition [11,12].

The diagnostics and monitoring rely on the history of pain and spasm of the masticatory muscles, which is frequently diagnosed subjectively by physician palpation. However, our understanding of bruxism could be improved by analyzing how changes in the features of the MMs are affected by different situations [13].

The US provides real-time and dynamic imaging of the MMs that mainly stabilizes the grinding path and is responsible for the bite forces during chewing. It gives information about muscle size, shape, and muscle movement during specific tasks or functional activities [13].

By combining SWE and US, comprehensive information about MM in patients with bruxism can be obtained. While quantitative evaluation provides biometric data such as size and shape, as well as stiffness, owing to dynamic imaging, structural and functional changes during jaw movements can be captured. Also, by tracking changes in muscle stiffness and size over time, it can be possible to monitor the treatment progress and the effectiveness of therapeutic approaches such as splints.

Masseter dysfunction in bruxism can manifest in two forms: atrophic or hypertrophic. While overuse can lead to hypertrophy in the early stages, persistent pain can result in disuse atrophy in chronic cases [14,15]. MM thickness can be used as an objective measure [15,16].

In our study, we obtained MM width and thickness values in resting and clenching positions. All the MMs were measured thicker and narrower in clenched status and thinner and wider in relaxed status. There was no difference between the right and left muscles. In terms of comparison of the MMs of painful bruxer and healthy control subjects, muscle thickness increased during contraction in both groups; however, it was thinner in the bruxers (10.97 ± 0.44 mm) than in the control group (11.98 ± 0.32 mm). During relaxation, these values were 9.17 ± 0.40 mm and 10.38 ± 0.27 mm, respectively. The thickness difference between the two groups was significant during relaxation but not during contraction. Both in relaxation and contraction, muscle width was significantly lower in bruxers (*p* < 0.05).

Quantifying the size of MM has limited value because of the difficulty of defining normalized dimensions due to the many variables that affect muscle size, such as age, gender, body mass index, and certain medical problems. However, studies have shown that there is a noticeable difference in muscle size, especially during relaxation, in bruxism patients compared to normal individuals, although a cut-off value could not be determined [15,17]. Jafari et al. [15] reported greater MM thickness in bruxers. However, Tetsuka et al. [16] and Toker et al. [17] measured thinner, narrower, and shorter MMs in the relaxed jaw.

We believe that symptomatology and duration of the disease would determine whether the change in the masticatory muscles will favor hypertrophy or atrophy. Our study cohort included painful bruxers with a VAS score of four or higher who had been suffering from this condition for at least 1 year. We concluded that lower muscle thickness and width measurements were due to the disuse of the MMs caused by chronic pain. Additionally, it may be useful to compare the thickness measurements over time in the same patient for monitoring the treatment.

MM stiffness has been explained by several hypotheses and theories. The main ones are relaxing deficiency caused by the inhibition of sarcoplasmic reticular calcium uptake, the over-release of acetylcholine leading to persistent contraction, and edematous changes due to masticatory myalgia [18,19].

SWE is a promising new imaging technique using ultrasound waves to measure the stiffness of tissues. It seems to have the potential to be an objective biomarker that can quantify masticatory muscle stiffness with high reproducibility, low operator dependency, strong reliability, and easy applicability [17,18,19].

As a consequence of different approaches to measurement, the values of shear wave elastography are expressed in two units: m/s and kPa. m/s represents the speed at which shear waves propagate in a given tissue, while kPa is calculated from the velocity of the shear waves using Young’s elastic modulus. Because the retrospective calculation of kPa from m/s and vice versa is not feasible, authors have recommend providing results using either kPa or both units [13]. We preferred to use kPa in our measurements.

Concerning the MMs, only a few studies have examined in vivo normative values. Among healthy people, the values at rest ranged from 5.25 kPa in the study by Takashima et al. [18] to 10.4 ± 3.7 kPa in the study by Arda et al. [20] and 10.0 ± 4.3 kPa in the study by Herman et al. [21]. The stiffness values of healthy volunteers were reported as 42.82 ± 5.56 kPa at rest and 53.36 ± 8.46 kPa during jaw clenching in the study by Ariji et al. [22]. In our study, the values of healthy subjects were 27.73 ± 1.92 kPa in relaxation and 35.52 ± 2.64 kPa in contraction status. The heterogeneity of study groups, devices, software, and measurement techniques could be responsible for this difference between the values.

In some studies, it was investigated whether there is a difference between the two masseter muscles. A significant correlation was observed between the right and left masseter stiffness index in healthy volunteers, indicating similar elasticity [19,21]. Olchowy et al. [13] and Costa et al. [23] reported that the MM elasticity index was higher in the symptomatic side in patients with temporomandibular disorders (TMDs) because of adopting a chronic unilateral chewing pattern in these patients. In our study group, there was no statistically significant difference between right and left MMs in terms of stiffness in either the bruxers or the healthy controls (*p* > 0.005).

Regarding the MM hardness in patients being greater than those of healthy volunteers, there are several quantitative studies in TMD, which is an umbrella term, but very few specifically focus on bruxism [18,22,23,24]. A systematic search was carried out with search engines that can access the MEDLINE database for studies of MM hardness measurements using SWE in patients with bruxism accompanied by myofascial pain. No result matched all the keywords. However, in a very recent study, Toker et al. [17] investigated the stiffness of MMs in 10 bruxism patients and 10 healthy controls. They reported a significantly higher velocity in bruxers than in healthy controls at the relaxed and closed jaw with values of 1.92 ± 0.44 m/s in bruxers and 1.66 ± 0.24 m/s in healthy participants. In our study, we included 26 MMs of painful bruxers with VAS scores ≥ 4 and 26 healthy MMs. We expressed the measurement of the muscle stiffness in kPa as recommended [13]. We obtained higher stiffness values in MMs of the painful bruxers than in the controls as 39.13 ± 4.52 kPa for bruxers and 27.73 ± 1.92 kPa for non-bruxers during relaxation. The difference between the two groups was statistically significant for MM stiffness during relaxation (*p* = 0.008) but not during contraction (*p* = 0.105).

A multidisciplinary approach for the treatment and a case-by-case approach for the management is recommended in bruxism patients. Treatment options include an array of methods, such as occlusal splints, intraoral devices, botulinum toxin A injections, medications, behavioral techniques, jaw exercises, and massage, as well as treating associated disorders such as sleep apnea and gastroesophageal reflux disease [1,25]. Bruxism may cause TMD associated with myofascial pain. In the treatment of muscle-related TMD and management of bruxism, physical therapy protocols and physical exercise, with or without radial extracorporeal shock wave therapy (rESWT), low-level laser therapy (LLLT), and laser acupuncture therapy (LAT), may be effective in reducing pain [26,27,28].

Daytime bruxism is predominantly characterized by tonic activity, contrary to sleep bruxism which is by phasic [29]. Because the nocturnal contractions are in the style of teeth grinding and brief, repeated contractions, the nocturnal wearing of splints is recommended to reduce symptoms like tooth wear and overloading of the joint. Occlusal splints are the most common treatment option and are effective in relieving symptoms as they decrease muscle activity [17,23,29].

We also investigated the relationship between the usage of occlusal splints and MM dimensions in painful bruxers. In comparison to users and non-users, there was no statistically significant difference in terms of thickness and width values between the two groups with *p*-values higher than 0.10. The effect of the occlusal splint on the diameters of the MM is not explicitly mentioned in the literature. However, there are some reports related to the effects of splints on asymmetry patterns of the masseter muscle [30]. Further research may be needed to explore this aspect in more detail.

Occlusal splints basically have an effect on muscle hyperactivity. By redistributing the forces, in addition to decreasing MM stiffness, they also help to reduce pain, improve jaw function, and prevent further damage to the teeth and jaw joints. Ispirgil et al. [31] investigated the hemodynamic effects of occlusal splint therapy on painful MMs of patients with bruxism accompanied by myofascial pain by using near-infrared spectroscopy. They found that splint usage caused a decrease in hyperemic response, which is indicative of a decrease in MM activity. However, there are contradictory studies about the effectiveness of occlusal splints in reducing MM stiffness and pain. Deregibus et al. [32] reported a randomized, controlled trial where occlusal splints did not have significant effects on reducing pain over a six-month period in TMD patients with a diagnosis of myofascial pain. Nevertheless, they emphasized that they noticed a trend in decreasing pain, but there was no control group undergoing splint therapy as a limitation. Olchowy et al. [33] showed reduced fatigue in the MMs of TMD patients after eight weeks of manual therapy and occlusal splint stabilization. They reported a significant decrease in stiffness and pain. In our study, regarding the MM stiffness measurements in occlusal splint users, both in relaxation and contraction, the mean values were lower than in non-users. But a significant difference between the two groups was obtained in relaxing state when the mean values were 26.91 ± 2.13 kPa for those wearing occlusal splints and 38.16 ± 3.61 kPa for those not wearing them. This difference was strongly significant with a *p*-value less than 0.01 (*p* = 0.006).

Our study has limitations, such as a relatively small sample size, lack of defined normalized values of the MMs, and the fact that the cohort represents only painful bruxism patients.

Finally, when measurements were taken during rest, significant differences were found in terms of stiffness and thickness in the MMs of bruxism patients with myofascial pain compared to normal individuals. The MMs were stiffer, narrower, and thinner in painful bruxers than healthy controls, as well as particularly stiffer in those not wearing occlusal splints than ones wearing them.

## 5. Conclusions

Bruxism with myofascial pain is a common masticatory dysfunction and requires an objective biomarker for diagnosis. The collaboration of SWE and dynamic B-mode US provides objective and quantitative data about the MM in myofascial pain patients with bruxism. So, it could be a potential instrumentally based assessment tool on STAB protocol. Measurements should be taken during the rest position because significant differences in stiffness and thickness became visible in the relaxation state. Bruxism patients with myofascial pain had significantly harder, narrower, and thinner MMs than healthy individuals. Also, stiffer MMs were measured in painful bruxers not wearing occlusal splints compared to those wearing them. These techniques can enhance our understanding of the condition, assist in diagnosis and treatment planning, and contribute to the development of personalized therapeutic approaches.

## Figures and Tables

**Figure 1 jpm-13-01467-f001:**
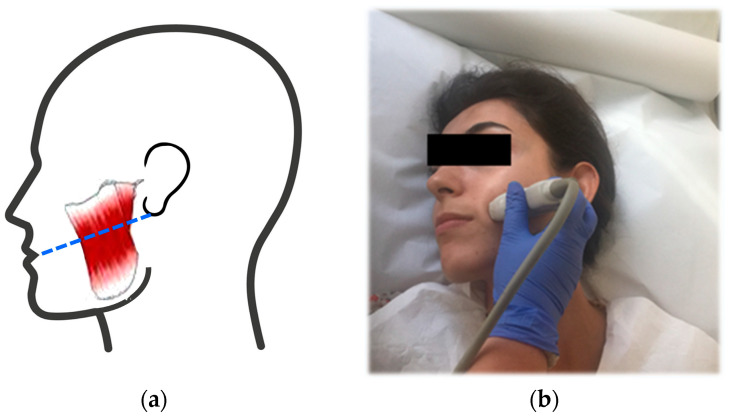
Scanning of the MMs. (**a**) The probe was placed on a line between the angle of the mouth and the lower implantation of the ear, crossing the bulky part (dashed line). (**b**) The subject was laid in supine position and her head was slightly turned in the opposite direction. A water-based transmission gel was applied. The transducer was held perpendicular and placed on the line.

**Figure 2 jpm-13-01467-f002:**
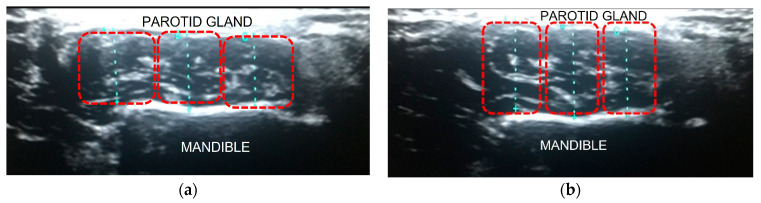
Dynamic B-mode US imaging; thickness measurement of the left MM. Note that the length of the blue dashed lines indicates the thickness of the muscle, and the red dashed boxes show anterior–middle–posterior thirds of the MM in order from left to right. (**a**) Thickness during muscle relaxation: 7.8 mm, 8.2 mm, and 7.6 mm of anterior, middle, and posterior thirds, respectively. (**b**) Thickness during contraction: 10.3 mm, 10.9 mm, and 9.8 mm in order of anterior to posterior thirds. Under each condition, the measurements were repeated three times and averaged.

**Figure 3 jpm-13-01467-f003:**
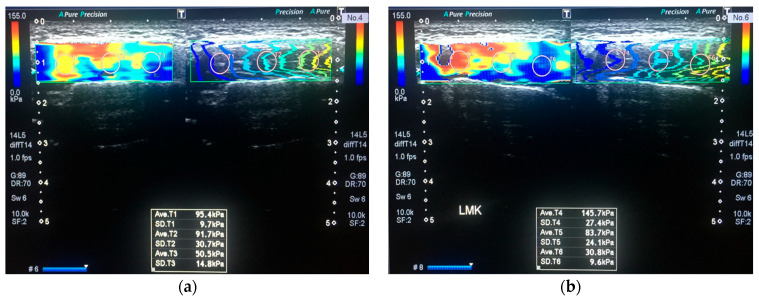
SWE of a 26-year-old woman with bruxism (VAS score is 5); dynamic imaging of the left MM. (**a**) The left MM in resting position on the shear wave propagation mode. Measurements were obtained from the ROI’s placed into regularly parallel contour lines. The E_max_ and E_mean_ values of the muscle on elasticity mode were 95.4 kPa and 79.2 kPa, respectively. (**b**) MM in clenching position with maximum force. The stiffness values were 145.7 kPa and 86.7 kPa in order of E_max_ and E_mean_.

**Figure 4 jpm-13-01467-f004:**
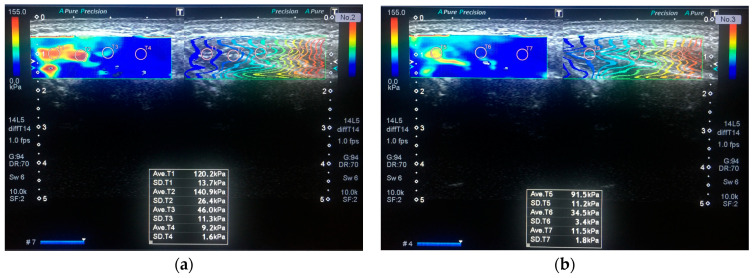
SWE on the propagation mode in the resting position. A 28-year-old woman with VAS scores of 4 in the right MM and 2 in the left MM. (**a**) The right MM. The E_max_ and E_mean_ values of the muscle on elasticity mode are 140.9 kPa and 79 kPa, respectively. (**b**) The left MM. The E_max_ is 91.5 kPa and E_mean_ is 45 kPa.

**Table 1 jpm-13-01467-t001:** Measurements of MM antero-posterior (thickness) and transverse (width) dimensions during relaxation and contraction (clenching) on dynamic B ultrasound.

Measurement (mm)	MM	Relaxation (Mean ± SD)	Contraction (Mean ± SD)	*p*-Value
Thickness	Right	10.16 ± 1.49	11.61 ± 1.85	0.000 *
(AP diameter)	Left	9.86 ± 1.89	11.73 ± 2.02	0.000 *
Width	Right	48.29 ± 1.45	44.15 ± 1.43	0.000 *
(TR diameter)	Left	48.84 ± 1.63	43.58 ± 1.52	0.000 *

* Statistically significant (*p <* 0.05), AP: antero-posterior, TR: transverse Student’s *t*-test.

**Table 2 jpm-13-01467-t002:** Comparison of thickness, width, and stiffness values of the MMs on relaxed and clenched positions in the painful bruxer and healthy control groups (Mean ± SD).

Masseter Muscle		Thickness (mm)		Width (mm)		Stiffness (kPa)	
Status		Relaxation (Mean ± SD)	Contraction (Mean ± SD)	Relaxation (Mean ± SD)	Contraction (Mean ± SD)	Relaxation (Mean ± SD)	Contraction (Mean ± SD)
Group A (*n* = 26)		9.17 ± 0.40	10.97 ± 0.44	44.08 ± 1.24	43.57 ± 1.52	39.13 ± 4.52	44.04 ± 4.99
Group B (*n* = 26)		10.38 ± 0.27	11.98 ± 0.32	50.55 ± 1.35	48.83 ± 1.64	27.73 ± 1.92	35.52 ± 2.64
Group A vs. Group B	Difference	1.21	1.01	6.47	5.26	11.40	8.52
	CI	0.23–2.18	0.13–2.14	2.78–10.17	0.77–9.76	3.07–19.75	1.84–18.87
	*p*-value	0.016 *	0.081 **	0.001 *	0.023 *	0.008 *	0.105 **

* Statistically significant (*p* < 0.05), ** statistically insignificant (*p* ≥ 0.05). Independent Samples *t*-test. CI: confidence interval Group A: MMs with VAS ≥ 4, Group B: healthy MMs.

**Table 3 jpm-13-01467-t003:** Comparison of thickness, width, and stiffness values of the MMs on relaxed and clenched positions according to occlusal splint usage in the painful bruxer group (Group A) (mean ± SD).

Masseter Muscle		Thickness (mm)		Width (mm)		Stiffness (kPa)	
Status		Relaxation (Mean ± SD)	Contraction (Mean ± SD)	Relaxation (Mean ± SD)	Contraction (Mean ± SD)	Relaxation (Mean ± SD)	Contraction (Mean ± SD)
Group AI (*n* = 14)		9.73 ± 1.41	11.28 ± 1.52	49.60 ± 8.13	44.81 ± 7.44	26.91± 2.13	37.21 ± 3.06
Group AII (*n* = 12)		10.45 ± 2.03	12.28 ± 0.52	46.90 ± 7.18	42.34 ± 7.48	38.16 ± 3.61	39.64 ± 4.03
Group AI vs. Group AII	Difference	0.72	1.0	2.70	2.48	11.25	2.43
	CI	0.34–1.78	0.21–2.2	−1.76–7.15	−1.79–6.75	3.40–19.12	7.63–12.50
	*p*-value	0.173 **	0.101 **	0.23 **	0.25 **	0.006 *	0.629 **

* Statistically significant (*p* < 0.05), ** statistically insignificant (*p* ≥ 0.05). Independent samples *t*-test. CI: confidence interval, SD: standard deviation. Group AI: MMs of splint user bruxers, Group AII: MMs of splint non-user bruxers.

## Data Availability

The data presented in this study are available on request from the corresponding author. The institutional board approved the use of data by the researcher for the purpose of this research, but uploading patient information to a public database was not included in this approval.

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
