# Peer review of "Dynamic Quantitative Imaging of the Masseter Muscles in Bruxism Patients with Myofascial Pain: Could It Be an Objective Biomarker?"

_jpm, 2023, doi:10.3390/jpm13101467_

Round 1

Reviewer 1 Report

Dear Authors,

This study aimed to investigate whether the collaboration of Shear Wave Elastosonography (SWE) and B-Mode Ultrasonography (US) could be offered as diagnostic tools to assess the presence, severity, and progress of bruxism, as well as a biomarker for the effectiveness of treatment in daily clinical practice.

The study is of scientific interest and in line with the aims of the Journal. However, there are some issues that should be added.

I suggest improving the abstract section. 

  • When start a sentence, in english language you should not use number. Please, write 52 as Fifty-two.
  • “52 masseter muscles (MMs) of 26 subjects (7 men and 19 women) were included inthe study”. This is Result information and should be reported after Methods. The same for line 176.

I suggest improving the introduction section. 

·      I suggest reporting myofascial pain as a TMD condition. Please refer to the Diagnostic Criteria for TMD (DC/TMD) Axis I. Thus, report that TMD could be divided in Group I: muscle disorders (including myofascial pain with and without mouth opening limitation); Group II: including disc displacement with or without reduction and mouth opening limitation; Group III: arthralgia, arthritis, and arthrosis.). (cite and refer to: Schiffman E, Ohrbach R, Truelove E, et al. Diagnostic Criteria for Temporomandibular Disorders (DC/TMD) for Clinical and Research Applications: recommendations of the International RDC/TMD Consortium Network* and Orofacial Pain Special Interest Group. J Oral Facial Pain Headache. 2014;28(1):6-27.).

·      Please report epidemiological data on bruxism in TMD patients  (Is bruxism associated with temporomandibular joint disorders? A systematic review and meta-analysis. Evid Based Dent. 2023 Jul 20. doi: 10.1038/s41432-023-00911-6).

Material and Methods

·      52 MMs of 26 subjects (7 men and 19 women) were enrolled in this study.” Please report this information in the Result section.

·      Line 97: please replace RDC/TMD with DC/TMD

·      Please remove the number of inlcuded patients from this section. You should only describe the methods in this section,

Result

·      In all the text, please use “P” or “p-value”.

Discussion

·      Well described. Please report more recent literature on conservative approaches and report new proposed treatment in the scientific literature (Effects of a Physical Therapy Protocol in Patients with Chronic Migraine and Temporomandibular Disorders: A Randomized, Single-Blinded, Clinical Trial. J Oral Facial Pain Headache. 2018 Spring;32(2):137-150. doi: 10.11607/ofph.1912. Effects of Radial Extracorporeal Shock Wave Therapy in Reducing Pain in Patients with Temporomandibular Disorders: A Pilot Randomized Controlled Trial. Applied Sciences. 2022; 12(8):3821. doi: 10.3390/app12083821. A randomized clinical trial comparing the efficacy of low-level laser therapy (LLLT) and laser acupuncture therapy (LAT) in patients with temporomandibular disorders. Lasers Med Sci. 2020 Feb;35(1):181-192. doi: 10.1007/s10103-019-02837-x. 

Author Response

Dear Sir, we appreciated for taking the time to review this manuscript. I would like to thank you for your valuable comments and point out the corrections we made in line with what you deemed necessary, which are indicated in red. Please find the detailed responses in the attached file. The corresponding revisions/corrections highlighted changes in the re-submitted files.

Sincerely.

Reviewer 2 Report

Point 1: The abstract and also section of method should be changed, with a correct description of the study. Study investigates a diagnostic tool in comparison on two lots: a study group of bruxers (with and without dental appliance treatment) and a control group of non-bruxers.

Method should be described accordingly, with specification of the diagnostic tool used for the assessment of masseter muscles.

Point 2: The authors should refer to the recent paper of Manfredini for bruxism diagnostic, and STAB tool for assessment of bruxism.

Manfredini D, Ahlberg J, Aarab G, Bender S, Bracci A, Cistulli PA, Conti PC, De Leeuw R, Durham J, Emodi-Perlman A, Ettlin D, Gallo LM, Häggman-Henrikson B, Hublin C, Kato T, Klasser G, Koutris M, Lavigne GJ, Paesani D, Peroz I, Svensson P, Wetselaar P, Lobbezoo F. Standardised Tool for the Assessment of Bruxism. J Oral Rehabil. 2023 Jan 3. doi: 10.1111/joor.13411. Epub ahead of print. PMID: 36597658.

Point 3: Revise all the phrases where you describe the study lot. Study lot is composed of people not masticatory muscles.

Point 4: Revise limitations: since you compared bruxers with non-bruxers…

Point 1: The abstract and also section of method should be changed, with a correct description of the study. Study investigates a diagnostic tool in comparison on two lots: a study group of bruxers (with and without dental appliance treatment) and a control group of non-bruxers.

Method should be described accordingly, with specification of the diagnostic tool used for the assessment of masseter muscles.

Point 2: The authors should refer to the recent paper of Manfredini for bruxism diagnostic, and STAB tool for assessment of bruxism.

Manfredini D, Ahlberg J, Aarab G, Bender S, Bracci A, Cistulli PA, Conti PC, De Leeuw R, Durham J, Emodi-Perlman A, Ettlin D, Gallo LM, Häggman-Henrikson B, Hublin C, Kato T, Klasser G, Koutris M, Lavigne GJ, Paesani D, Peroz I, Svensson P, Wetselaar P, Lobbezoo F. Standardised Tool for the Assessment of Bruxism. J Oral Rehabil. 2023 Jan 3. doi: 10.1111/joor.13411. Epub ahead of print. PMID: 36597658.

Point 3: Revise all the phrases where you describe the study lot. Study lot is composed of people not masticatory muscles.

Point 4: Revise limitations: since you compared bruxers with non-bruxers…

Author Response

(The authors gave the same response as above.)
